# Combined Effect of Midazolam and Bone Morphogenetic Protein-2 for Differentiation Induction from C2C12 Myoblast Cells to Osteoblasts

**DOI:** 10.3390/pharmaceutics12030218

**Published:** 2020-03-02

**Authors:** Yukihiko Hidaka, Risako Chiba-Ohkuma, Takeo Karakida, Kazuo Onuma, Ryuji Yamamoto, Keiko Fujii-Abe, Mari M. Saito, Yasuo Yamakoshi, Hiroshi Kawahara

**Affiliations:** 1Department of Dental Anesthesiology, School of Dental Medicine, Tsurumi University, 2-1-3 Tsurumi, Tsurumi-ku, Yokohama 230-8501, Japan; 2911002@stu.tsurumi-u.ac.jp (Y.H.); fujii-keiko@tsurumi-u.ac.jp (K.F.-A.); kawahara-h@tsurumi-u.ac.jp (H.K.); 2Department of Biochemistry and Molecular Biology, School of Dental Medicine, Tsurumi University, 2-1-3 Tsurumi, Tsurumi-ku, Yokohama 230-8501, Japan; chiba-r@tsurumi-u.ac.jp (R.C.-O.); karakida-t@tsurumi-u.ac.jp (T.K.); yamamoto-rj@tsurumi-u.ac.jp (R.Y.); saito-mari@tsurumi-u.ac.jp (M.M.S.); 3National Institute of Advanced Industrial Science & Technology, Central 6, 1-1-1 Higashi, Tsukuba, Ibaraki 305-8566, Japan; espkaz009@gmail.com

**Keywords:** drug repositioning, cell, hydroxyapatite, bone, midazolam

## Abstract

In drug repositioning research, a new concept in drug discovery and new therapeutic opportunities have been identified for existing drugs. Midazolam (MDZ) is an anesthetic inducer used for general anesthesia. Here, we demonstrate the combined effects of bone morphogenetic protein-2 (BMP-2) and MDZ on osteogenic differentiation. An immortalized mouse myoblast cell line (C2C12 cell) was cultured in the combination of BMP-2 and MDZ (BMP-2+MDZ). The differentiation and signal transduction of C2C12 cells into osteoblasts were investigated at biological, immunohistochemical, and genetic cell levels. Mineralized nodules formed in C2C12 cells were characterized at the crystal engineering level. BMP-2+MDZ treatment decreased the myotube cell formation of C2C12 cells, and enhanced alkaline phosphatase activity and expression levels of osteoblastic differentiation marker genes. The precipitated nodules consisted of randomly oriented hydroxyapatite nanorods and nanoparticles. BMP-2+MDZ treatment reduced the immunostaining for both α1 and γ2 subunits antigens on the gamma-aminobutyric acid type A (GABAA) receptor in C2C12 cells, but enhanced that for BMP signal transducers. Our investigation showed that BMP-2+MDZ has a strong ability to induce the differentiation of C2C12 cells into osteoblasts and has the potential for drug repositioning in bone regeneration.

## 1. Introduction

Drug repositioning is a research method that searches for new drug effects using existing drugs for which the safety and pharmacokinetics in humans have already been confirmed, increasing the practical applications. The greatest advantages of drug repositioning are certainty about the safety and pharmacokinetics at the clinical level and low cost, since existing data can be used. In this study, we were motivated to use mesenchymal stem cells (MSCs) to discover the potential for drug repositioning for future bone treatments and organ regenerative medicine.

MSCs are undifferentiated pluripotent progenitor cells capable of self-renewal and differentiation [1]. In research on osteogenic differentiation using MSCs, differentiation from MSCs to osteoblasts is regulated by many molecular factors and mechanical stimuli. Human mesenchymal stem cells (hMSCs) differentiate into osteoprogenitor cells, preosteoblasts, osteoblasts, and osteocytes under appropriate culture conditions, and express several important osteogenic genes during this differentiation process [2]. In vitro differentiation of MSCs depends on culture conditions, and growth factors, such as those of the transforming growth factor β (TGF-β) family, significantly affect MSC differentiation [3,4].

C2C12 cells are an immortalized mouse skeletal muscle cell line [5] and have been widely used as model cells for undifferentiated mesenchymal cells to study the differentiation of myoblasts, osteoblasts, and myogenesis; to express various target proteins; and to investigate mechanistic biochemical pathways. In vitro studies using C2C12 cells demonstrated that bone morphogenetic protein-2 (BMP-2) converts the developmental pathway of C2C12 from a myogenic cell lineage to an osteoblastic cell lineage [6]. Furthermore, the in vitro mineralization of C2C12 cells is induced by the presence of dexamethasone, ascorbic acid, β-glycerol phosphate, all-trans retinoic acid (ATRA), and 2-*N*,6-*O*-sulfated chitosan [7,8,9].

Midazolam (MDZ) is a chemically synthesized imidazobenzodiazepine derivative with pharmacological effects, including hypnotic, sedative, anesthetic, anxiolytic, muscle relaxant, and anticonvulsant effects [10]. Intravenous MDZ preparations have been recommended as first-line drugs for the treatment of status epilepticus [11]; however, intravenous MDZ products have not been approved for the treatment of status epilepticus in most countries, but have been used off-label for patients as a first-line drug for this condition in Japan. In research using MSCs, MDZ negatively affects the cell viability and osteogenic differentiation in human bone marrow-derived mesenchymal stem cells [12]. In several in vitro studies investigating the effects of MDZ on tumor and cancer cells, MDZ was found to induce cellular apoptosis by regulating the caspase pathway, endoplasmic reticulum stress, autophagy, and the cell cycle [13,14,15,16]. In the dental field, MDZ enhances the differentiation of a porcine dental pulp-derived cell line to odontoblasts and promotes the formation of dentin-like hydroxyapatite [17]. However, little is known about the combined effect of MDZ and growth factors on cells.

Here, we examined the combined effects of BMP-2 and MDZ on osteogenic differentiation of C2C12 cells at the biological, immunohistochemical, crystal engineering, and genetic cell levels.

## 2. Materials and Methods

Our studies received approval from the Institutional Ethics Review Committee of the Tsurumi University School of Dental Medicine (Yokohama, Japan). All animal experiments were approved by the Institutional Animal Care Committee and the Recombination DNA Experiment and Biosafety Committee of the Tsurumi University School of Dental Medicine (Project identification code #1318, 1 December 2015). All experiments were performed in accordance with relevant guidelines and regulations.

### 2.1. Alkaline Phosphatase (ALP) Activity Assay

The mouse myoblast cell line, C2C12 [18], was obtained from the RIKEN Cell Bank (Tsukuba Science City, Ibaraki, Japan). The cells were plated on a 96-well plate at a density of 1.0 × 10^4^ cells/well and were cultured with the standard medium consisted of alpha Minimum Essential Medium (αMEM; Thermo Fisher Scientific, Waltham, MA, USA) containing 10% fetal bovine serum (FBS), 50 U/mL Penicillin and 50 μg/mL Streptomycin (Thermo Fisher Scientific, Waltham, MA, USA) in a humidified 5% CO_2_ atmosphere for 24 h at 37 °C. The medium was changed to a growth medium supplemented with 0, 2.5, 5, 10, 20, or 40 μM MDZ (Merck, Darmstadt, Germany) with 125, 250, or 500 ng/mL recombinant human (rh)BMP-2 (#355-BEC, R&D Systems, Minneapolis, MN, USA) with or without 50 nM LDN-193189 (Tocris Bioscience, Bristol, UK). The ALP activity measurement process in each well was described previously [19]. After 72 additional hours of incubation, the cells were washed once with phosphate-buffered saline (PBS), and ALP activity was assayed using 10 mM p-nitrophenylphosphate as the substrate in a 100 mM 2-amino-2-methyl-1,3-propanediol-HCl buffer (pH 10.0) containing 5 mM MgCl_2_ and incubated for 10 min at 37 °C. We added 0.2 M NaOH to quench the reaction, and the absorbance at 405 nm was read on a plate reader.

### 2.2. Cell Proliferation Assay

The cells were plated on 96-well plates at a density of 1.0 × 10^4^ cells/well for cell proliferation assay (MTS assay) or 1.0 × 10^3^ cells/well for the measurement of cell population doubling level in standard medium and cultured at 37 °C in a humidified 5% CO_2_ atmosphere. The culture medium was changed every other day. The proliferation rate of the cells on six 96-well plates was determined on days 1, 2, and 3 using a CellTiter 96^®^AQuous One Solution Cell Proliferation Assay (MTS assay) (Promega Corporation, Madison, WI, USA).

### 2.3. Immunofluorescent Staining of Myosin Filaments

C2C12 cells were grown on a 12-well plate at an initial density of 1.0 × 10^5^ cells/cm^2^. After incubation for 24 h, the medium was changed to a mineralization-inducing medium containing 10 mM β-glycerophosphate and 50 μM ascorbic acid (calcification medium) supplemented with or without 20 μM of MDZ and/or 500 ng/mL recombinant human bone morphogenetic protein-2 (rhBMP-2). The cells were cultured for an additional 10 days, changing half the amount of calcification medium every other day, and cells were fixed with 4% paraformaldehyde for 30 min at room temperature. The cells were permeabilized with 0.1% Triton X for 5 min and incubated in a blocking solution (1% BSA, 10% normal goat serum) for 1 h at room temperature. For primary antibody application, the dilution of an anti-fast myosin skeletal heavy chain antibody (MY-32; #ab51263, Abcam, Cambridge, UK) was used at 1:400 dilution, and the cells were incubated overnight at 4 °C. For secondary antibody application, a diluted Alexa Fluor^TM^ 488 rabbit anti-mouse immunoglobulin G (IgG) heavy and light chains (H+L) cross-adsorbed secondary antibody (Thermo Fisher Scientific, Waltham, MA, USA), was used at 1:1000, and the cells were incubated for 2.5 h in darkness protected from light at room temperature. Fluorescent micrographs were obtained using a fluorescence microscope (Biozero BZ-8100, Keyence, Osaka, Japan).

### 2.4. Quantitative Polymerase Chain Reaction (qPCR) Analysis

The cells were extracted with an RNA extraction kit (Roche Diagnostics GmbH, Mannheim, Germany). After the purified total RNA (2 μg) was reverse transcribed, the reaction mixture consisted of SYBR Green PCR master mix (Roche Diagnostics GmbH) supplemented with 0.5 µM forward and reverse primers and 2 µL of cDNA as template. The specific primer sets were designed using Primer-BLAST as a primer designing tool [20]. The specific primer sets and running conditions are shown in Appendix A. Glyceraldehyde-3-phosphate dehydrogenase (*Gapdh*) was used as the reference gene. Each ratio was normalized to the relative quantification data of runt-related transcription factor 2 (*Runx2*), osterix (*Osx*), tissue nonspecific alkaline phosphatase (*Tnsalp*), and myoblast determination protein 1 (*MyoD*) in comparison to the *Gapdh*, which was generated on the basis of a mathematical model for relative quantification in qPCR system.

### 2.5. Detection of Mineralized Nodules

C2C12 cells were grown on a 12-well plate at an initial density of 1.0 × 10^5^ cells/cm^2^. After incubation for 24 h, the medium was changed to a calcification medium supplemented with or without 20 μM of MDZ and/or 500 ng/mL rhBMP-2. The cells were cultured for up to 10 days. Mineralization was visualized using Alizarin red S staining. After fixation with 4% paraformaldehyde neutral buffer solution for 30 min, the cells were stained with 1% Alizarin red S (Sigma-Aldrich, St. Louis, MO, USA) solution for 10 min, then washed with distilled water, and photographed.

C2C12 cells were grown on a 24-well plate at an initial density of 1.0 × 10^5^ cells/cm^2^. After incubation for 24 h, the medium was changed to a calcification medium supplemented with 0, 5, 10, or 20 μM MDZ in the presence or absence of 500 ng/mL rhBMP-2. The cells were cultured for up to 10 days. Each well on the plates was rinsed with PBS, and the calcium was dissolved in 0.5 mL of 0.5 N HCl by gentle rocking for 30 min. The calcium concentration in the eluate was spectrophotometrically determined at 570 nm by following the color development with a calcium assay kit (Calcium C-test Wako, Wako Pure Chemical Industries, Ltd., Osaka, Japan).

### 2.6. X-ray Diffraction (XRD) Measurement

The precipitates containing calcium salts and cells were crushed in agate mortar with 99.5% ethanol, and an aliquot was placed on a silicon non-reflective plate and naturally dried for XRD measurement. The measurements were performed for four types of samples: precipitate after three weeks of cell culture, after two weeks of cell culture, and after three weeks of cell culture without β-glycerophosphate, and pure midazolam reagent as a reference.

An X-ray diffractometer (RINT 2000, Rigaku, Akishima, Tokyo, Japan) with monochromated Cu-Kα radiation was used at 40 kV and 200 mA for characterization. Two types of scanning were used for measurements: continuous scanning at 2*θ* from 3 to 60° at a rate of 1°/min and step scanning at 2*θ* from 25 to 35° with a step of 0.02°. A 30 s data accumulation per step was employed in the step scanning mode.

The diffraction peak positions for calcium salts were compared to those of three calcium phosphate phases that were referred to in the Joint Committee on Powder Diffraction Standards (JCPDS) cards (dicalcium phosphate dihydrate (DCPD) (CaHPO_4_): card 11–293, octacalcium phosphate (OCP) (Ca_8_(HPO_4_)_2_(PO_4_)_4_·5H_2_O): card 26–1056, hydroxyapatite (HAP) (Ca_10_(PO_4_)_6_(OH)_2_): card 9–432).

### 2.7. Transmission Electron Microscope (TEM) Observation and Elemental Analysis using Scanning-TEM Energy-Dispersive X-ray Spectroscopy (STEM-EDS)

The crushed samples containing calcium salts and cells after three weeks of cell culture were placed on a Cu grid for TEM observation with an acceleration voltage of 200 kV. Macroscopic characterization for the major types of calcium phosphates that consisted of calcium salts was performed using selected area electron diffraction (SAED) patterns obtained at an 800 nm ϕ area. Microscopic characterization for each crystal phase was performed by analysis of the fast Fourier transform (FFT) pattern of the high-resolution TEM (HR-TEM) image of crystal.

An elemental analysis of calcium salts was performed using a Super-X EDS system in the TEM (STEM-EDS) (National Institute of Advanced Industrial Science & Technology, Tsukuba Science City, Ibaraki, Japan). To minimize electron damage against samples, the probe diameter, beam amplitude, and beam residence time at each position were set to ~0.5 nm, ~0.55 nA, and 10 μs, respectively. The whole analysis of the measurement area was completed within 5 min. The average Ca/P atomic % ratio and the contained minor elements were clarified for the calcium salts.

### 2.8. Immunostaining of GABAA Receptor, Phosphorylated-Smad (p-Smad)1/5/8, and the Type I BMP Receptor

C2C12 cells on chamber slides were fixed with 4% paraformaldehyde for 15 min at room temperature and incubated in a blocking solution (1% BSA, 10% normal goat serum) for 1 h at room temperature. For primary antibody application, the dilution values of the anti-polyclonal antibodies were 1:500 for GABAA receptor α1 (GABAARα1) (#ab33299; Abcam, Cambridge, UK), 1:1000 for GABAA receptor γ2 (GABAARγ2) (#224003; Synaptic Systems, Goettingen, Germany), and 1:100 for both phosphorylated-Smad1/5/8 (p-Smad1/5/8) (#9511; Cell Signaling, Danvers, MA, USA) and BMP receptor 1A (#ab38560; Abcam, Cambridge, UK). The cells were incubated overnight at 4 °C. For secondary antibody application, a diluted HRP-conjugated goat anti-rabbit IgG H+L antibody (Abcam, Cambridge, UK) was used at 1:500, and the cells were incubated for 1 h at room temperature. The positive signal was detected using 3,3-diaminobenzidine (DAB; TaKaRa, Kusatsu, Japan) as a staining substrate. Sections were counterstained using hematoxylin to clearly observe tissue and cell morphology. Light micrographs were obtained using a Canon EOS Kiss X8i camera (Canon, Tokyo, Japan) on an optical microscope (OLYMPUS BX50, Olympus, Tokyo, Japan). The positive rate of cells for p-Smad1/5/8 antibody was calculated using Image J software Version 1.52a (National Institutes of Health, Bethesda, MD, USA).

### 2.9. Statistical Analysis

For the ALP assay, the MTS assay, and the qPCR and calcium analyses, all values are presented as the mean ± standard error of the mean (SEM). Statistical significance was determined using the Mann-Whitney U test for the ALP assay, calcium analyses and cell positive rate, the nonparametric Steel’s test for the MTS assay, and the Steel-Dwass test for the qPCR. In all cases, *p* < 0.01 or *p* < 0.05 were regarded as statistically significant.

## 3. Results

### 3.1. Differentiation of the C2C12 Cells

MDZ is a short-acting benzodiazepine derivative with a molecular weight of 325.77 g/mol (Figure 1a). We first attempted to find the optimum concentration of MDZ and BMP-2 for the differentiation induction of C2C12 myoblast cells to osteoblasts. Since ALP is used as the initial marker for the differentiation of mesenchymal cells into hard tissue-forming cells such as osteoblasts [6], we investigated the effects of MDZ and BMP-2 on ALP activity in the C2C12 cells. With 500 ng/mL rhBMP-2, used in our previous study [8], we initially explored the optimal concentration of MDZ in a range from 0.1 to 100 μM and found that 10 μM MDZ has the highest ALP activity in C2C12 cells (Appendix A). We next narrowed the concentration range of MDZ (0–40 μM) and sought the optimum concentration for the combination of MDZ and rhBMP-2 (125–500 ng/mL; Figure 1b). When the concentration of rhBMP-2 was used at 125 or 250 ng/mL, ALP activity did not increase at any concentration of MDZ. The use of 500 ng/mL rhBMP-2 dramatically increased the ALP activity level depending on the concentration of MDZ; the ALP activity at 10 and 20 μM MDZ in particular was approximately 3.0-fold higher than that at 0 µM MDZ. However, the use of 40 μM MDZ significantly reduced the activity. Using 500 ng/mL rhBMP-2 and/or 20 μM MDZ as an optimum concentration, we evaluated the osteoblast differentiation ability of C2C12 cells by measuring ALP activity (Figure 1c). The addition of MDZ alone (MDZ) did not increase ALP activity and it was at the control level (Cont), but when rhBMP-2 was added (rhBMP-2), an increase in ALP activity was observed. Furthermore, when rhBMP-2 and MDZ were used in combination (rhBMP2+MDZ), the ALP activity was increased about 3-fold as compared to rhBMP-2 alone. We next performed the same experiment in the presence of 50 nM LDN-193189, which is a selective BMP type I receptor inhibitor. In the group where rhBMP-2 was added, the ALP activity remarkably reduced to the same level. We interpret these findings to indicate that osteogenic differentiation of C2C12 cells is mainly induced by BMP-2, and that MDZ alone does not possess the osteogenic differentiation potential of C2C12 cells. Thus, our interesting finding is that MDZ had the effect of promoting the action of BMP-2.

### 3.2. Cytotoxicity of MDZ for Cell Proliferation Rate of C2C12 Cells and Cultured Cells after Confluence

Since the use of 20 μM and 500 ng/mL as optimum concentrations for MDZ and rhBMP-2, respectively, was more effective at enhancing ALP activity, we investigated the cytotoxicity of MDZ (20 μM) for the cell proliferation rate in C2C12 cells and for the cultured cells after confluent (Figure 2a). Under all conditions, the cells reached confluence in the first day, and the cell proliferation rate decreased with the passage of days after confluence. We found no significant differences in cell proliferation rates (Figure 2b) or the cell population doubling level (Figure 2c) among all conditions for three days. In particular, MDZ (20 μM) did not affect cell proliferation of C2C12 cells or the cultured cells after confluence. We interpret these results to indicate that MDZ (20 μM) does not cause any cytotoxicity in our experiments using C2C12 cells.

### 3.3. Immunofluorescent Staining of Myosin Filaments in C2C12 Cells 

Based on the above result that MDZ and BMP-2 do not affect the cell proliferation of C2C12 cells and the cultured cells after confluent, we next immunohistologically investigated the direction of differentiation of C2C12 cells in the presence of MDZ and BMP-2 (Figure 3). When we cultured the cells in the absence of MDZ and rhBMP-2 (i.e., control), we observed bundles of myosin filaments extending for long distances through the myotubes. Myosin filaments were also observed in the cells cultured with MDZ alone, but those were somewhat immature compared to the control. In contrast, no myotubes were observed in cells cultured in rhBMP-2 alone or rhBMP-2+MDZ. Thus, we found that MDZ alone does not have a beneficial effect on C2C12 cells but possesses some auxiliary effect for the action of BMP-2 on the differentiation of C2C12 cells to other cells.

### 3.4. Gene Expression in the C2C12 Cells

We next investigated the effect of MDZ and BMP-2 on gene expression in C2C12 cells. The gene expression of a panel of osteoblastic markers and muscle differentiation regulators in C2C12 cells at days 1 and 3 after MDZ and/or rhBMP-2 treatment was analyzed using qPCR (Figure 4). For osteoblastic markers, we quantified the mRNA expression levels of *Runx2*, *Osx*, and *Tnsalp*. On day 1, the expression levels of *Runx2*, *Osx*, and *Tnsalp* in cells cultured with rhBMP-2 or rhBMP-2+MDZ were significantly higher (1.93–2.37-fold for *Runx2*; 28.9–36.7-fold for *Osx*) (*Tnsalp* could not be calculated because the mRNA level of the control was not determined) than in cells cultured without MDZ or rhBMP-2 (i.e., control). Even compared to MDZ alone, the levels were significantly increased (1.54–1.89-fold for *Runx2*; 56.3–71.5-fold for *Osx*). In contrast, on day 3, the expression levels of *Runx2* and *Osx* remarkably decreased, but the mRNA level of *Tnsalp* in cells cultured with rhBMP-2+MDZ dramatically increased (approximately 2200-fold) compared to that of the control. We also amplified *MyoD* as a myogenic marker. On day 1, the mRNA levels of *MyoD* in cells cultured with rhBMP-2 and rhBMP-2+MDZ were not significantly different. On day 3, the culture with rhBMP-2 significantly reduced the expression level of *MyoD*. In contrast, the mRNA level of *MyoD* in cells cultured with MDZ was significantly higher than in cells cultured with rhBMP-2 or rhBMP-2+MDZ on days 1 and 3, although its level tended to decrease on day 3.

### 3.5. Mineralization Induction in C2C12 Cells

To further investigate the effect of MDZ and BMP-2 on mineralization inducibility, we cultured C2C12 cells in a mineralization-inducing culture medium. The nodule formation and mineralization capacities of the cells were assessed with Alizarin red S staining (Figure 5a). Ten days following mineralization induction, no mineralized nodules were observed in the plate of the cells cultured in mineralization-inducing culture medium with or without MDZ alone or rhBMP-2 alone. In contrast, the plate of the cells cultured in mineralization-inducing culture medium with rhBMP-2+MDZ displayed mineralized nodules with Alizarin red S staining.

We also quantitatively analyzed the calcium content in C2C12 cells cultured in mineralization-inducing culture medium in the presence or absence of rhBMP-2 (Figure 5b). Ten days following mineralization induction, the cells cultured in the rhBMP-2+MDZ displayed a dramatically increased amount of calcium deposition.

### 3.6. XRD Patterns

Based on the above results, we focused on characterizing the mineralized nodules from C2C12 cells at the crystal engineering level. Figure 6a shows XRD patterns for three types of samples. The pattern corresponding to the sample after three weeks of cell culture (blue curve, BM-3 weeks) consisted of two sharp and intense peaks at 2*θ* = 28.2° (Peak 1) and 40.5° (Peak 3), and a low intense broad peak at 2*θ* = 31.8° (Peak 2) that tailed to a higher 2*θ*. An intense but broad peak was observed at approximately 2*θ* = 20.0°, and broad and low intense peak was observed at approximately 2*θ* = 8.0°.

Intensities of the above peaks drastically decreased, except that of the peak at 2*θ* = 20.0° in the sample after two weeks of cell culture (magenta curve, BM-2 weeks). The pattern obtained by the step scanning mode (Figure 6a, inset small panel) showed faint peaks that corresponded to Peaks 1 and 2.

In the pattern corresponding to the sample cultured without β-glycerophosphate (green curve, M sample) showed a low intense peak of 2*θ* = 20.0°. Peak 1 was still observed in the step scanning mode (Figure 6a, inset small panel); however, Peak 2 disappeared. Since the lack of β-glycerophosphate inhibits the precipitation of calcium phosphates, Peak 2 was attributed to the calcium phosphates. Both HAP and OCP have the several intense peaks at 2*θ* = 31.0–33.0°, but DCPD does not (Appendix A). The crystallinity of precipitates was low based on the broad peak width. Since the partially disordered OCP along [100] does not show the most intense peak at 2*θ* = 4.7° [21], Peak 2 likely corresponded to the peak of HAP and/or OCP. DCPD, OCP, and HAP do not have peaks at 2*θ* = 28.2 (Peak 1) and 40.5° (Peak 3); therefore, these two peaks were attributed to the material other than calcium phosphates.

Figure 6b shows the pattern of midazolam powder. The crystalline midazolam showed an intense peak at peak 1 and a less intense peak at peak 3. Since peak 1 was observed for the M sample, we concluded that both peaks 1 and 3 are attributed to midazolam.

### 3.7. TEM Images of Calcium Phosphate Nodules

Figure 7a shows a TEM image of calcium phosphate nodules. The macroscopic morphology of the nodule was a sphere with ~300 nm diameter that consisted of flexible nanofibers (<10 nm width) and thin plates. This morphology was the same as that of calcium phosphates precipitated by osteoblastic cells without midazolam [22].

The SAED pattern corresponding to this nodule showed two Debye rings (Figure 7b): an evident ring at 3.571 nm^−1^ from the center, and a faint ring at 2.924 nm^−1^ from the center. These two rings corresponded to the interplanar distances, *d*, of 0.280 nm (outer) and 0.342 nm (inner), respectively. The planes corresponding to *d* = 0.280 nm were {211}, {112}, and {121} of HAP; {420}, {710}, {511}, etc. of OCP; and {002} of DCPD, within an error of 1.5%. The planes corresponding to *d* = 0.342 nm were {002} of HAP and {002}, {121}, etc. of OCP, within error.

High-resolution (HR)-TEM image of the thin plate region in Figure 7a revealed lattice fringes (Figure 7c). The corresponding FFT image (Figure 7d) showed periodicities for two directions, with magenta and light blue arrows. The *d* of the magenta arrow direction was 0.811 nm that corresponds to {100} and {110} of HAP (<1.5% error). OCP and DCPD did not have corresponding planes. The *d* of 0.811 nm corresponded to the minimum distance of the fringes observed in the right half area in Figure 7c. The *d* of the light blue arrow direction was 0.554 nm, and the intersecting angle between magenta and light blue was 71.3°. No corresponding planes had this *d* or intersected to [100] or [110] at 71.3°. The FFT image corresponding to the lower left area in Figure 7c showed the diffraction spots that are the same as those in the light blue arrow direction in Figure 7d (Figure 7e). Light blue spots were attributed to this area despite the partly disordered lattice fringes. Among DCDP, OCP, and HAP, the *d* of 0.554 nm only corresponded to {111} and {111} planes of OCP. This means that the precipitates were a mixture of HAP and OCP.

The HR-TEM image of nanofibers and its FFT analysis showed a *d* of 0.345 nm, which corresponded to the {002} plane of HAP or OCP (Appendix A). Since HAP frequently shows a morphology elongated to [001], these fibers were attributed to HAP.

### 3.8. STEM-EDS Analysis for Calcium Phosphate Nodules

Figure 8 shows a high-angle annular dark-field (HAADF) image (Figure 8a) and two-dimensional elemental mappings (Figure 8b–d) for the precipitate shown in Figure 6a. Detection of the N atom indicated the contamination of organic materials such as proteins in the nodule. The EDS spectrum of the nodule in Figure 6a showed the Ca/P atomic % ratio as 1.40 ± 0.01, and Na and Mg were detected as minor elements. S, Cl, and K were also detected, but their amounts were very small (Figure 8e). We measured five different precipitates that showed the Ca/P ratio of 1.38–1.42.

### 3.9. Immunohistochemical Study for Signal Transduction in C2C12 Cells

We further attempted to gain information about the effect of MDZ and BMP-2 on signal transduction in C2C12 cells. The C2C12 cells cultured in mineralization-inducing culture media with or without rhBMP-2 and MDZ for seven days (Figure 9a) and one day (Figure 9b) were observed in hematoxylin and eosin (H&E)-stained sections. The same C2C12 cells were used for an immunohistochemical study using anti-polyclonal antibodies against mouse GABAARα1 and GABAARγ2, respectively and BMP receptor type 1A (BMPR1A; Figure 9a), and phosphorylated- Smad1/5/8 (p-Smad1/5/8; Figure 9b). Compared to the control culturing cells in the absence of MDZ and rhBMP-2 (Control), the combination of rhBMP-2 and MDZ (rhBMP-2+MDZ) remarkably reduced the specific immunostaining for both GABAARα1 and GABAARγ2, but rhBMP-2+MDZ dramatically enhanced the specific immunostaining for BMP receptor type 1A and phosphorylated- Smad1/5/8. The positive rate of cells for p-Smad1/5/8 antibody was 3.93% for control, 16.9% for rhBMP-2 alone, 7.54% for MDZ alone, and 35.3% for rhBMP-2+MDZ (Figure 9c).

## 4. Discussion

BMP-2 is necessary for converting the differentiation pathway of C2C12 cells into osteoblast lineage [6]. To investigate the osteoblast differentiation in C2C12 cells, the measurement of ALP activity is used as a marker because the proliferating osteoblasts exhibit its activity. An in vitro study using hMSCs showed that MDZ possesses a suppressive effect on osteogenic differentiation because it inhibits ALP activity and calcium deposition in hMSCs [12]. The above study; however, did not consider the action of BMP-2, so we examined the combined effect of MDZ and BMP-2. We demonstrated that the rhBMP-2+MDZ dramatically enhanced ALP activity over rhBMP-2 alone. This finding suggests that MDZ alone is not involved in osteoblast differentiation, but it can be used in combination with rhBMP-2 to promote the action of BMP-2.

In neuroblastoma cell lines, human leukemia cells, and colon cancer cells, MDZ activates caspase-9, caspase-3, and poly(ADP-ribose) polymerase, indicating the induction of the mitochondrial intrinsic pathway of apoptosis [15,16]. In human oral squamous cell carcinoma (OSCC) cell lines, MDZ induces necrotic cell death rather than apoptosis [14]. We demonstrated that the use of 40 μM MDZ reduces ALP activity, suggesting the cytotoxicity of MDZ; but 20 μM MDZ did not affect ALP activity, cell proliferation, or the cell population doubling level of C2C12 cells. Thus, our results suggest that MDZ possesses different sensitivities to target cells and that the effect varies with dose.

In the development of skeletal muscle, mesenchymal stem cells differentiate into myoblasts, and myoblasts fuse with each other to differentiate into multinucleated myotubes and eventually mature into muscle fibers with contractile ability. Myoblasts produce dynamic morphological changes during muscle differentiation, such as the degradation and reconstruction of actin filaments, cell membrane fusion, and organelle reorganization. A group of genes called muscle regulatory factors (MRFs) play an important role in the differentiation of myoblasts, including myogenic factor 5, MyoD, myogenin, and myogenic regulatory factor 4 [23]. In addition to the action of MRF genes, other factors, such as folic acid, promote the myogenic differentiation of C2C12 cells [24]. In immunostaining with a myosin heavy chain antibody, we demonstrated that MDZ alone is prone to driving C2C12 cells to myotube formation, although the action of MDZ promoting the myogenic differentiation is unclear. Surprisingly, we found that MDZ leads C2C12 cells to the differentiation of osteoblast lineage, promoting the action of BMP-2. However, how MDZ promotes the action of BMP-2 is still unknown.

Osteoblast culture models containing C2C12 cells have been well characterized for identifying the temporal sequence of gene and protein expression related to osteoblast differentiation. The osteogenic differentiation of mesenchymal stem cells in vitro has been generally classified into three stages: osteoprogenitor cells, pre-osteoblasts, and mature osteoblasts [25]. Those osteoblastic cell differentiations are mainly controlled by the transcription factor Runx2 [26] and osterix [27]. BMP-2 is known to enhance the transcriptional activity of Runx2 [28] and regulate osterix through Msx2 and Runx2 during osteoblast differentiation [29]. The osteoblastic cell differentiation at the initial stage has been characterized by the transcription and protein expression of ALP [30]. In particular, the expression of ALP is found in preosteoblasts. We demonstrated that the mRNA levels of *Runx2*, *Osx*, and *Tnsalp* dramatically increased in C2C12 cells in the presence of rhBMP-2. In contrast, *MyoD*, a marker gene for the muscle differentiation, increased in C2C12 cells in the presence of MDZ alone, but its level hardly increased in the presence of rhBMP-2. These results suggest that the C2C12 cells used in this study may have differentiated into preosteoblast-like cells. One notable finding is that rhBMP-2+MDZ enhanced mRNA levels of *Runx2* on day 1 and of *Osx* and *Tnsalp* on day 3. These findings suggest that MDZ produces an osteoblast differentiation effect in the presence of BMP-2.

Alizarin red S staining has been widely used to evaluate calcium deposits in cell culture [31]. The mineralization of C2C12 cells can be detected by the treatment with 800 ng/mL rhBMP-2 alone [9]. In the present experiment, because the ED_50_ of the rhBMP-2 we used, was 80–480 ng/mL, we used 500 ng/mL. In the case of the observation of mineralized nodules in C2C12 cells by Alizarin red S staining, most experiments have been performed for cells cultured in the calcification medium for more than 12 days. Our previous study has revealed that the addition of ATRA requires the induction of mineralization [8]. In the present study, the number of culture days was 10 days, and ATRA was not added. We demonstrated that no mineralized nodules were observed in C2C12 cells under the addition of rhBMP-2 alone, but that the combined use of rhBMP-2 and MDZ resulted in the evident formation of mineralized nodules. This finding suggests that the mineralized nodules possess calcium-phosphate-based components and that rhBMP-2+MDZ is the most effective at inducing mineralization in C2C12 cells. We further revealed that the major portion of mineralized nodules was HAP, with a small volume of OCP. As shown by the less intense Debye rings and damage by electron bombardment in TEM observation, the crystallinity of the precipitate was essentially low, which is consistent with the broad Peak 2 in the XRD pattern. The initial calcium phosphate phase precipitated in this system would be amorphous calcium phosphate (ACP), as discussed in previous studies [17]. ACP can transform into both OCP and HAP; however, OCP is a transient phase in this route and easily changes to HAP. In this reaction process, the OCP structure is unstable toward [100] orientation and frequently assumes a HAP-like structure [21]; therefore, it easily changes to HAP. Despite the lack of the strongest diffraction peak of OCP, 2*θ* = 4.7° in XRD, we conclude that OCP crystals formed in this system. This conclusion is consistent with the biological analysis that showed preosteoblast-like cells. The strong and sharp Peak 1 indicates that the midazolam was well crystallized; however, the most intense peak of midazolam, which should appear at 2*θ* of ~20.0°, was not observed in the sample after three weeks of cell culture. Instead, a broad and highly intense peak appeared. Although the diffraction from cells shows a peak at this angle [17], our data were not attributed to an independent cell peak, since amorphous materials such as cells do not produce intense peaks, compared to the crystalline material, midazolam. The amorphous state of midazolam, as a potential origin of this peak, is also contradicted by the other sharp peaks of this material. A reasonable explanation for the peak at 2*θ* of ~20.0° is that the midazolam was incorporated into the cells, which resulted in an intense but broad peak pattern.

GABAA receptors are heteropentamers composed of combinations of α, β, γ, δ, ε, π, and θ subunits [32]. In the central nervous system, the combination of α1, β2, and γ2 (α1:β2:γ2 = 2:2:1) is most commonly expressed, and binding sites for many drugs, such as γ-aminobutyric acid, benzodiazepine, and barbituric acid, are formed at the subunit boundaries [33]. The sensitivity to a drug varies depending on the subunits constituting the GABAA receptor, and GABAA receptors containing α1, α2, α3, and α5 subunits are sensitive to benzodiazepines, but those containing α4 and α6 subunits are insensitive [34,35]. MDZ binds to the benzodiazepine binding site at the boundary between α1 and γ2, and increases the action of GABA, which enhances the permeability of chloride ions. We demonstrated that the MDZ not only affected the localization of GABAA receptors in C2C12 cells, but rhBMP-2+MDZ clearly reduced the staining for both α1 and γ2 subunit antigens on the GABAA receptor. These findings suggest that the complex formed by the binding of MDZ to rhBMP-2 reduced the antibody reactivity by covering the epitope of the GABAA receptor and/or decreased the gene expression of the GABAA receptor. 

The BMP receptor constitutes a tetramer receptor with two molecules of type I and two molecules of type II, and the signal transduction occurs by binding to a ligand [36]. In the BMP-Smad pathway, a BMP ligand bound to types I and II receptors induces intracellular kinase activity of the receptors and phosphorylates Smad1, Smad5, and Smad8/9. These Smads are translocated to the nucleus by forming a complex with Smad4 to regulate the transcriptional activity [37]. Our immunohistochemical study revealed that rhBMP-2+MDZ enhanced the staining for both the BMP type I receptor and the phosphorylated Smad1/5/8 antigens in C2C12 cells cultured in mineralization-inducing culture media. These findings suggest that rhBMP-2+MDZ promotes the gene expression of the BMP type I receptor to combine more BMP-2 and consequently enhance the phosphorylation of Smad1/5/8.

An in vitro study showed that the Mg ions on the underlying substrates of the calcium phosphate cement facilitate the recognition of the BMP receptors and enhance signal transduction [38]. Our EDS analysis demonstrated that Mg ions are also contained in the mineralized nodules next to Ca and P. As rhBMP-2+MDZ enhanced the staining for phosphorylated Smad1/5/8 antigens in C2C12 cells cultured in mineralization-inducing culture media, our findings suggest that MDZ facilitates the recognition of the BMP receptors for BMP-2 binding as well as the Mg ion.

## 5. Conclusions

Besides the use of MDZ as an anesthetic inducer and sedative, MDZ has been shown in the dental field to enhance the differentiation of dental pulp cells to odontoblasts and promote the formation of dentin-like hydroxyapatite [17]. This study demonstrated its efficacy, and the combination with TGF-β or BMP-2 showed a tendency to be suppressed. We demonstrated that the combination of MDZ and BMP-2 promoted the differentiation of C2C12 cells into osteoblasts rather than MDZ alone. This finding seems to indicate that MDZ promotes TGF-β and BMP signaling in preference to GABAA receptor responsiveness in cells related to hard tissue formation. Notably, with the action of MDZ, some cells promote differentiation with the use of MDZ alone, and others facilitate differentiation while enhancing the action of BMP, as in this study. Thus, we identified the possibility of the drug repositioning of MDZ in bone regeneration research and clinical applications. However, the efficacy of MDZ in vivo has not yet been demonstrated. Further studies are required to clarify the pharmacokinetic and pharmacological efficacy of MDZ in animal experiments.

## Figures and Tables

**Figure 1 pharmaceutics-12-00218-f001:**
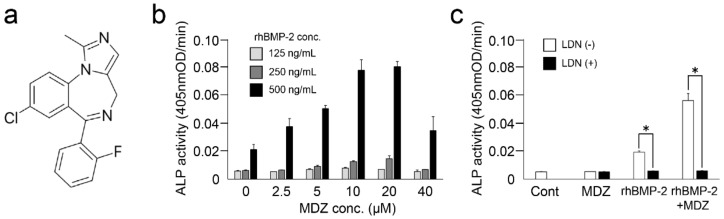
Combined effect of midazolam (MDZ) and bone morphogenetic protein-2 (BMP-2) on alkaline phosphatase (ALP) activity in C2C12 cells: (**a**) Structure of MDZ; (**b**) ALP-inducing activity of MDZ (0, 2.5, 5, 10, 20, and 40 μM) with recombinant human (rh)BMP-2 (125, 250, and 500 ng/mL); (**c**) inhibition experiment of ALP-inducing activity in the absence of rhBMP-2 and MDZ (Cont) or presence of 20 μM MDZ alone (MDZ), 500 ng/mL rhBMP-2 alone (rhBMP-2), and the combination of rhBMP-2 and MDZ (rhBMP-2+MDZ) containing 50 nM LDN-193189 (LDN). Values are the means ± standard error of the mean (SEM) of 6 culture wells (* *p* < 0.01, Mann–Whitney U test).

**Figure 2 pharmaceutics-12-00218-f002:**
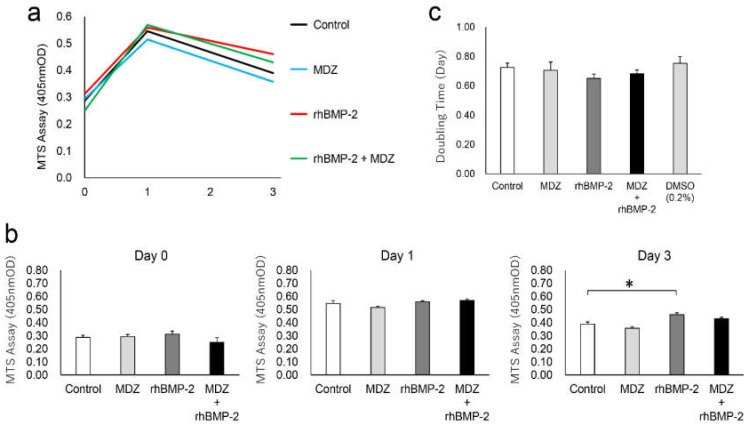
Cytotoxicity of MDZ for cell proliferation rate of C2C12 cells and for cultured cells after confluence incubated for different periods. (**a**) Changes in cell proliferation over time by MTS assay; (**b**) Comparison of cell proliferation on days 0, 1, and 3. C2C12 cells in the absence (Control) or presence of MDZ alone (MDZ), rhBMP-2 alone (rhBMP-2), and the combination of rhBMP-2 and MDZ (rhBMP-2+MDZ) on days 0, 1, and 3 were cultured at a final volume of 120 μL/well for 1 h at 37 °C. MTS reagent was added, and an absorbance of 450 nm was recorded using a microplate reader (*n* = 8 tests per sample). Values are the means ± SEM (* *p* < 0.01, Steel’s test). (**c**) Cell population doubling level against days.

**Figure 3 pharmaceutics-12-00218-f003:**
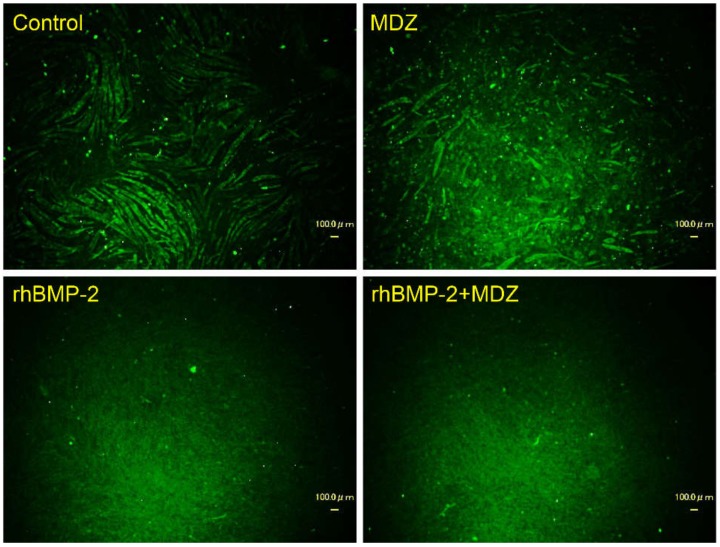
Combined effect of MDZ and BMP-2 on the differentiation of C2C12 cells. Fluorescence detection of Alexa Fluor™ 488-labeled myosin skeletal heavy chain in C2C12 cells on day 10 following culture in the absence (Control) or presence of MDZ alone (MDZ), rhBMP-2 alone (rhBMP-2), and the combination of rhBMP-2 and MDZ (rhBMP-2+MDZ). Myosin filaments (green) in fixed C2C12 cells were detected using an anti-fast myosin skeletal heavy chain antibody at a dilution of 1:400 and an Alexa Fluor^TM^ 488 rabbit anti-mouse immunoglobulin G (IgG) heavy and light chains (H+L) cross-adsorbed secondary antibody at a dilution of 1:1000 (scale bar = 100 μm).

**Figure 4 pharmaceutics-12-00218-f004:**
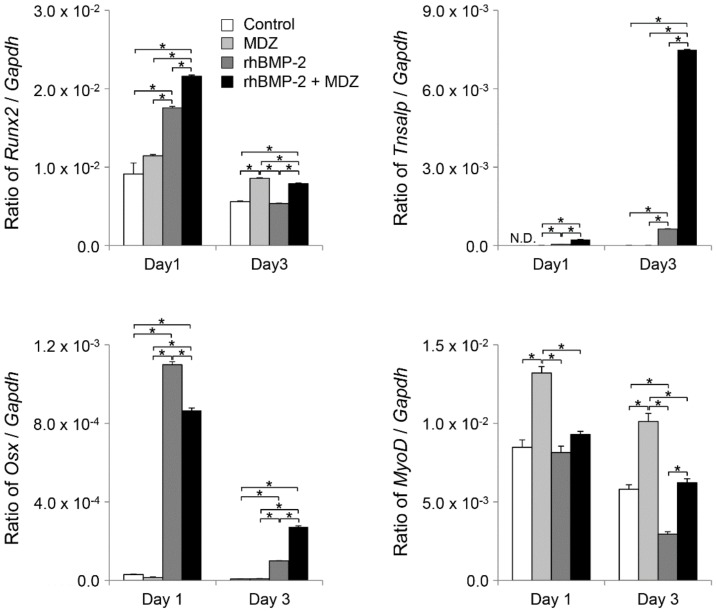
Combined effect of MDZ and BMP-2 on gene expression in the C2C12 cells. qPCR analysis on days 1 and 3 after culture in the absence (Control) or presence of MDZ alone (MDZ), rhBMP-2 alone (rhBMP-2), and the combination of rhBMP-2 and MDZ (rhBMP-2+MDZ). *Runx2*, runt-related transcription factor 2; *Osx*, osterix; *Tnsalp*, alkaline phosphatase; *MyoD*, myoblast determination protein 1; ND, not determined. Each mRNA expression value was normalized to that of the reference gene glyceraldehyde-3-phosphate dehydrogenase (*Gapdh*), and the relative quantification data for *Runx2*, *Osx*, *Tnsalp*, and *MyoD* in C2C12 cells were generated on the basis of a mathematical model for relative quantification in a qPCR system (*n* = 6). All values are presented as the mean ± SEM (* *p* < 0.05, Steel–Dwass test).

**Figure 5 pharmaceutics-12-00218-f005:**
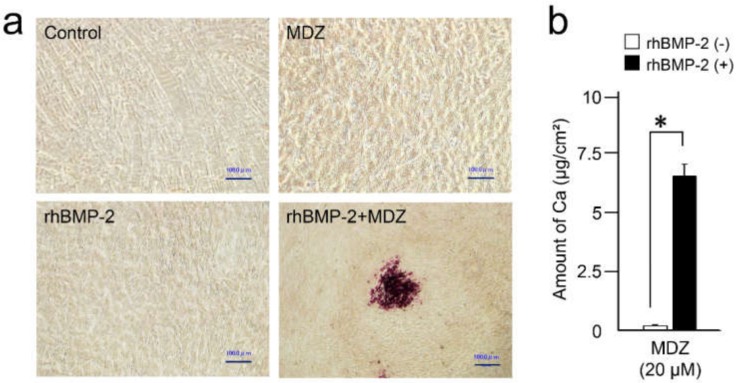
Combined effect of MDZ and BMP-2 on the mineralized nodule formation in the C2C12 cells. (**a**) Alizarin red S staining for nodule precipitates on day 10. C2C12 cells cultured in mineralization-inducing culture media in the absence (Control) or presence of MDZ alone (MDZ), rhBMP-2 alone (rhBMP-2), and the combination of rhBMP-2 and MDZ (rhBMP-2+MDZ). The rhBMP-2+MDZ exhibited nodule formation (scale bar = 100 μm). (**b**) Calcium contents in C2C12 cells were determined on day 10 after the mineralization induction. The amount of calcium increased in C2C12 cells cultured in mineralization-inducing culture media with 10 or 20 μM of MDZ in the presence of rhBMP-2. Values are the means ± SEM of 12 culture wells. Asterisk (*) indicates a significant difference (*p* < 0.01, Mann-Whitney U test) between the cells incubated with and without rhBMP-2.

**Figure 6 pharmaceutics-12-00218-f006:**
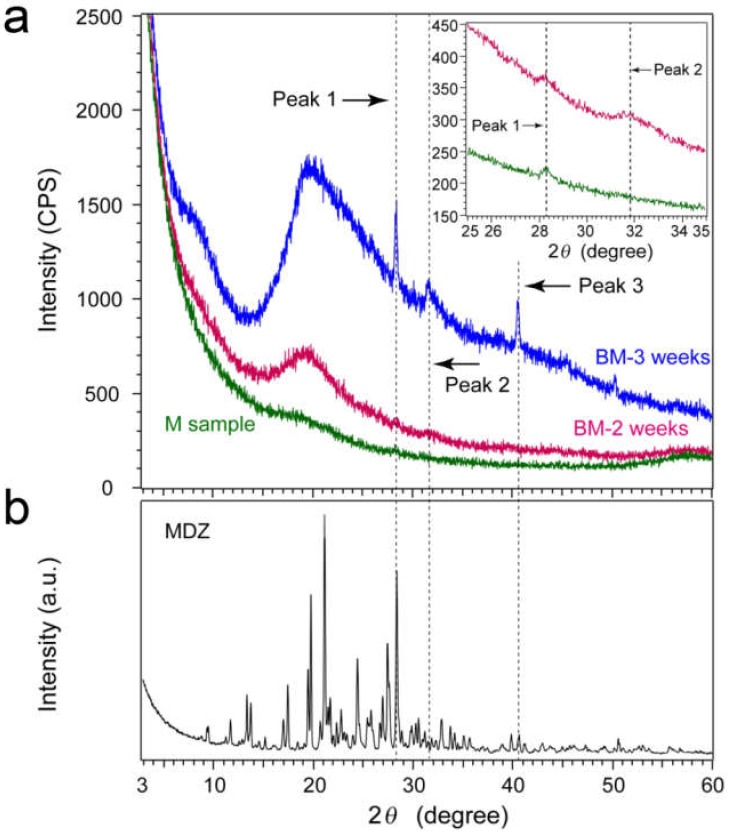
(**a**) Continuous-scanning XRD patterns of sample after three weeks of cell culture (blue), two weeks of cell culture (magenta), and without β-glycerophosphate (green). Inset small panel corresponds to patterns for blue and magenta curves measured by step scanning mode. (**b**) Continuous-scanning XRD pattern of midazolam.

**Figure 7 pharmaceutics-12-00218-f007:**
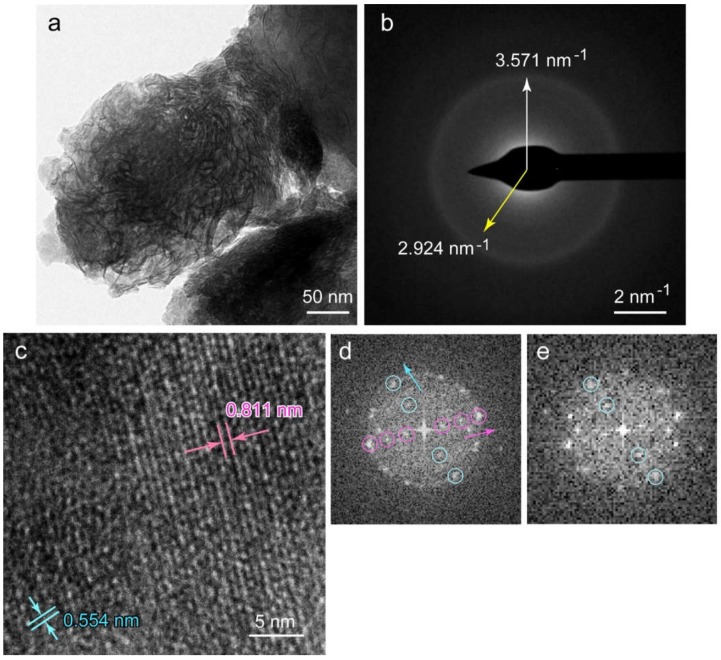
(**a**) Macroscopic TEM image of the calcium phosphate nodule. (**b**) Selected area electron diffraction (SAED) pattern of (**a**). (**c**) High-resolution (HR)-TEM image of the thin plate area in (**a**). (**d**) Fast Fourier transform (FFT) image for Figure 7c. Magenta arrow direction correspond to hydroxyapatite (HAP) [100], but the blue one is not consistent with any HAP orientations. (**e**) FFT image of lower left area in Figure 7c. The *d* corresponds to that of octacalcium phosphate (OCP) {111} and {111} planes.

**Figure 8 pharmaceutics-12-00218-f008:**
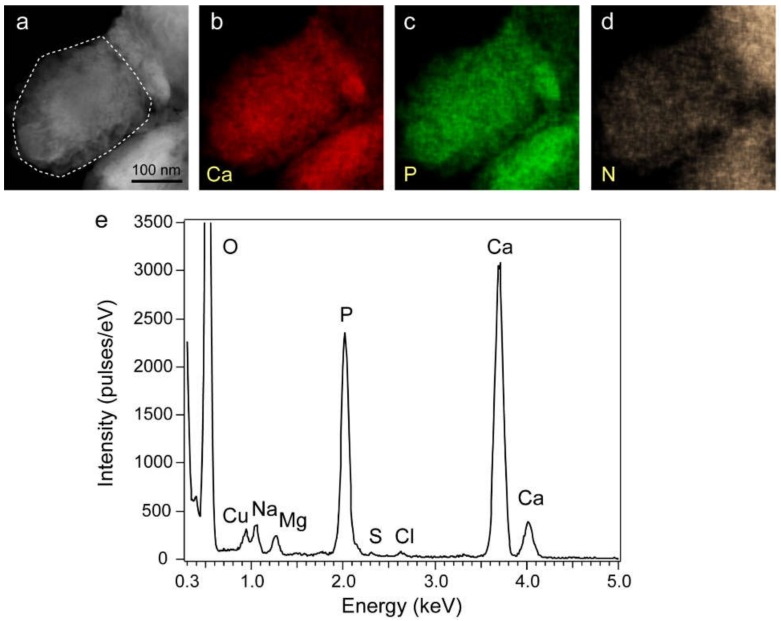
(**a**) HAADF image and (**b**–**d**) elemental mapping of the calcium phosphate nodule in Figure 7a. (**e**) Energy-Dispersive X-ray Spectroscopy (EDS) spectrum for white dotted area in (**a**).

**Figure 9 pharmaceutics-12-00218-f009:**
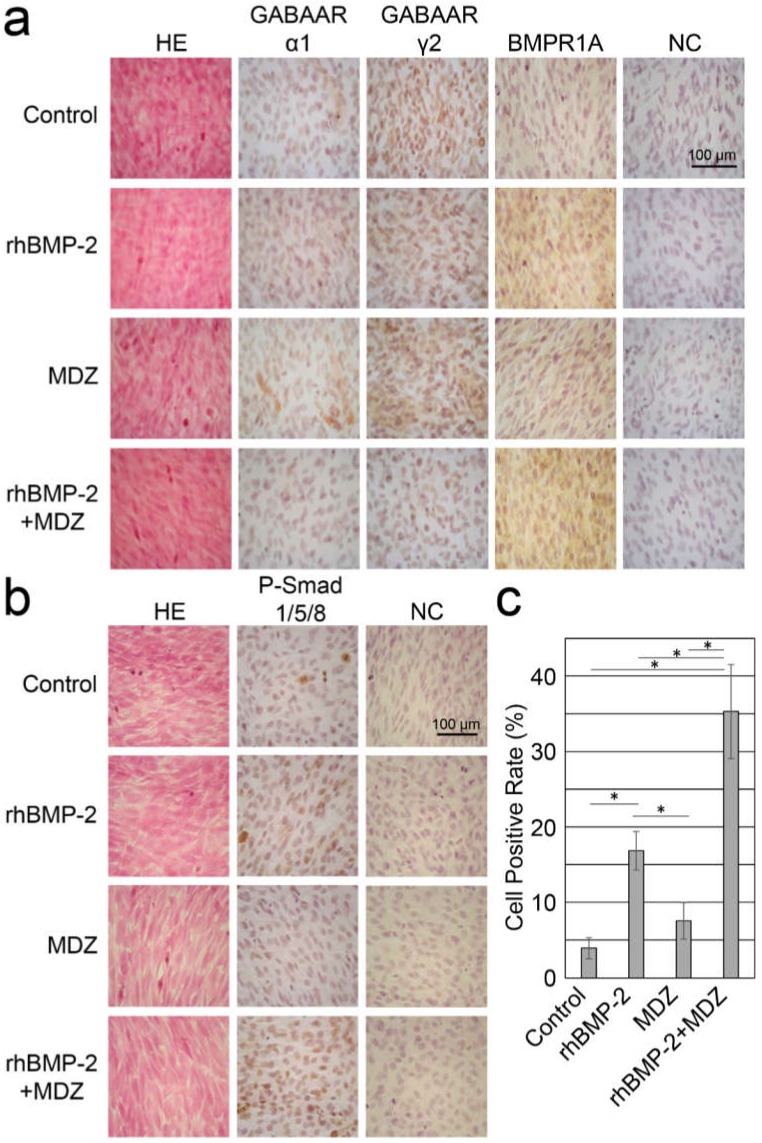
Combined effect of MDZ and BMP-2 on signal transduction in C2C12 cells. (**a**,**b**) Hematoxylin and eosin (H&E)-stained C2C12 cells detected by transmitted-light microscopy (magnification: 400×). The control was processed without a primary antibody (NC). Immunohistochemical detection of (**a**) GABAA receptor α1 (GABAARα1) and γ2 (GABAARγ2) subunits, and BMP receptor type 1A (BMPR1A) in C2C12 cells on day 7 and (**b**) phosphorylated-Smad1/5/8 (p-Smad1/5/8) in C2C12 cells on day 1 following culture in the absence (Control) or presence of rhBMP-2, MDZ, and rhBMP-2+MDZ. (**c**) Cell positive rate for p-Smad1/5/8 antibody (*n* = 5). Values are the mean percentage ± standard error (* *p* < 0.01, Mann-Whitney U test).

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
