# Peer review of "Combined Effect of Midazolam and Bone Morphogenetic Protein-2 for Differentiation Induction from C2C12 Myoblast Cells to Osteoblasts"

_pharmaceutics, 2020, doi:10.3390/pharmaceutics12030218_

Round 1

Reviewer 1 Report

The manuscript entitled "Combined Effect of Midazolam and Bone Morphogenetic Protein-2  for Differentiation Induction from C2C12 Myoblast Cells to Osteoblasts" by  Hidaka et al. addresses the hypothesis that midazolam, a well-known anesthetic drug in combination with the factor BMP-2 would modulate the differentiation of myoblast C2C12 cell line into a osteoblast. The introduction is concise, logic and clear. The results presented shows the experiments necessary to answer the hypothesis, are clear, follow a good logic of presentation and are quite impressive. The discussion is also clear and directly to the point, giving an strong message to the readers. 

I have no further comment to the Authors, but congratulate them for the interesting work.

Author Response

Thank you very much for taking the time to review our manuscript. We hope to continue this research and produce good results in the future.

Reviewer 2 Report

Excellent manuscript. Very novel information on rx medazoline and bmp

Author Response

(The authors gave the same response as above.)

Reviewer 3 Report

This manuscript assessed the combine osteogenic effects of BMP2 and MDZ drugs on C2C12 cells. This viewpoint is interesting and well-description. Overall, I think the osteogenic differentiation of C2C12 cells is mostly determined by BMP2. The real effect of MDZ drugs on C2C12 cells is useless, with only a little cell cytotoxicity. Thus, the combine effects of MDZ drugs on osteogenic differentiation of C2C12 cells is due to cell cytotoxicity. In addition, the real effect of MDZ drugs on C2C12 cells is seldom explained.      

Results: In the 3.1. differentiation of the c and Fig. 1C, it seems that the osteogenic differentiation of C2C12 cells are mostly induced by BMP2 rather than MDZ drugs, especially in the use of LDN-193189 inhibitor. Without BMP2, the capacities of MDZ drugs on osteogenic differentiation of C2C12 cells are almost useless. Please explain this finding. In addition, please add the osteogenic differentiation effect when only using the MDZ drugs. Results: In the Fig. 2, the proliferation of C2C12 cells decrease at the 2nd and 3rd day, which mean cells start to apoptosis or death. The really proliferative effects of BMP2 and MDZ drugs on C2C12 cells would be wrong in this experimental design. I think this result is not correct. Results: Immunofluorescent Staining of Myosin Filaments in C2C12 Cells. The Fig. 3 also revealed no any beneficial effect of MDZ drugs on C2C12 cells. Results: In the Fig. 5a, the mineralization of C2C12 cells with treatment of rhBMP-2 only should be detected from the previous publication. Results: In the Fig. 7 and 8, can it show the data from control experimental group. Results: In the Fig. 9, please add the quantitative measurements in this experiment.

Author Response

Please see the response in the file attached here.

Round 2

Reviewer 3 Report

Agree and accept